# Synergistic stabilization of a double mutant in chymotrypsin inhibitor 2 from a library screen in *E. coli*

Louise Hamborg[1,2], Daniele Granata[1], Johan G. Olsen[1], Jennifer Virginia Roche[1], Lasse Ebdrup Pedersen[2], Alex Toftgaard Nielsen [2], Kresten Lindorff-Larsen [1] & Kaare Teilum [1✉]

Most single point mutations destabilize folded proteins. Mutations that stabilize a protein typically only have a small effect and multiple mutations are often needed to substantially increase the stability. Multiple point mutations may act synergistically on the stability, and it is often not straightforward to predict their combined effect from the individual contributions. Here, we have applied an efficient in-cell assay in *E. coli* to select variants of the barley chymotrypsin inhibitor 2 with increased stability. We find two variants that are more than 3.8 kJ mol$^{-1}$ more stable than the wild-type. In one case, the increased stability is the effect of the single substitution D55G. The other case is a double mutant, L49I/I57V, which is 5.1 kJ mol$^{-1}$ more stable than the sum of the effects of the individual mutations. In addition to demonstrating the strength of our selection system for finding stabilizing mutations, our work also demonstrate how subtle conformational effects may modulate stability.

[1] Structural Biology and NMR Laboratory and the Linderstrøm-Lang Centre for Protein Science, Department of Biology, University of Copenhagen, Copenhagen N, Denmark. [2] The Novo Nordisk Foundation Center for Biosustainability, Technical University of Denmark, Kemitorvet, Lyngby, Denmark. ✉email: kaare.teilum@bio.ku.dk

Understanding how the stability of a protein changes when an amino acid residue is changed is fundamental for several biological processes and the aetiology of many diseases[1]. For using proteins in biotechnological and bio-pharmaceutical applications it is often an advantage that the proteins have long shelf-lives and are not degraded too rapidly during the applications[2]. Our ability to engineer proteins with increased stability or to understand how amino acid changes cause decreased stability has thus been the subject of large number of studies. Although our understanding of the physics and thermodynamics of protein stability is rather advanced and thus to a great extent allows us to calculate structures with amazing precision using either molecular dynamics simulations based on a physical a description of the interactions stabilizing a protein structure[3] or by bioinformatics and machine learning[4], we are still not able to accurately predict how multiple mutations in a protein act together to change the stability of the protein.

Recently, we described a system based on recombinant expression in *E. coli*, that can be used to measure both protein translation and folding stability in vivo[5]. The translation sensor part of the system is based on an RNA hairpin structure inserted into a polycistronic mRNA coding for the protein of interest and for the fluorescent protein mCherry, thus making it possible to read out efficient translation through a red fluorescent signal. The protein folding and stability sensor is based on GFP-ASV through a system engineered to be expressed as a response to protein misfolding. Green fluorescence is thus used as a proxy for low in vivo protein stability. This mis-folding response relies on a heat shock promoter, *lbpAp*, and the *E. coli* heat shock system. With increasing levels of protein misfolding, more of the chaperone DnaK will bind the misfolded protein instead of the *E. coli* heat shock sigma factor, RpoH. In the absence of DnaK, RpoH can participate in assembly of the RNA polymerase sigma 32 complex, which can drive transcription from the *lbpAp* promoter. The presence of misfolded protein that can bind DnaK thus results in the expression of GFP and a green fluorescence signal. By expressing libraries of random mutations in a given protein in this bacterial sensor system and analysing the cells by fluorescence-activated cell sorting (FACS), it is possible to select large sets of protein variants that retain a folded structure, thus avoiding complications from using a functional assay as a proxy for folding stability.

An alternative to screening mutant libraries for proteins with altered stability is to calculate the effect of substituting amino acids and find variants with the desired properties. Several computational tools have been developed that predict the change in free energy for folding ($\Delta\Delta G_f$) between a wild-type protein and a mutant[6–22]. In general, the methods perform rather well when predicting the effects of destabilizing mutations but often fail in predicting stabilizing mutations[23]. A comparison of several sta-bility predictors showed an average correlation of around 0.6 between experimentally determined and computed changes in stability for all types of mutations[24,25]. The algorithms are better at predicting deletion mutations in the hydrophobic core, than mutations that increase the size of the side chain, mutations on the protein surface and mutations where electrostatic interactions contribute to the stabilization. This is partly a result of the data available for training the algorithms that mainly consist of dele-tion mutations in the hydrophobic core[26], A particular challenge in predicting stabilizing protein variants is that among the few single substitutions that are actually stabilizing the effects are often small, so that multiple substitutions may be needed to create a substantial stabilizing effect[17]. As the effects of the mutations are not always independent, and non-additivity may result in both positive or negative epistasis[27,28], it can be difficult to predict the stability of proteins with multiple substitutions. One way to improve the computational methods is to generate stability data on a larger set of protein variants generated to scan sequence space better than the current available datasets and including also stabilizing variants.

Here, we have applied the bacterial sensor with the aim of selecting variants from a library of random mutations of barley chymotrypsin inhibitor 2 (CI2) to broadly cover sequence and stability space. CI2 is a small single domain protein of 64 residues, which has been extensively used as a model to understand key concepts of protein folding and stability[29–33]. CI2 is a highly stable protein with free energy for folding, $\Delta G_f = 31$ kJ mol$^{-1}$ at 25 °C, and heat unfolds at 79 °C[32,34]. Although CI2 has been extensively studied by mutagenesis, only few variants that stabilize the protein relative to the wild-type are reported and all of these are substitu-tions of R48 with a hydrophobic residue[35]. Extending this set of stabilized CI2 variants could, in the context of the extensive amount of data available for CI2, provide additional insight into how multiple mutations may interact to stabilize a protein.

In the current work we show that our sensor system can indeed be used to select for proteins with increased stability, even in an already highly stable protein. Many of the 25 variants for which we measured the stability destabilize CI2. Still, we found two variants, L49I/I57V and D55G, that are significantly stabilized relative to wild-type CI2. For L49I/I57V we find that there is a strong positive synergistic effect between the two substitutions and this variant is stabilized by 5.1 kJ mol$^{-1}$ more than the sum of the individual effects of the two single variants. A detailed analysis of the structural changes in L49I/I57V suggests that several subtle long-range effects underlie the high stability gain.

## Results

**FACS sorting libraries of random CI2 variants.** When wild-type CI2 is expressed in the bacterial sensor system only little GFP is produced (Fig. 1a). The dynamic range of the GFP signal towards discovering stabilized variants (i.e. less GFP) is thus very small. During the development of the folding sensor, we compared the GFP signal from wild-type CI2 with that from the highly stabi-lized variant R48I and found only a marginal difference[5]. It will thus not be possible to separate stabilized variants from the wild-type or from variants with stabilities close to that of the wild-type. A library of random mutations in wild-type CI2 will therefore be most suited for selecting variants with stabilities that are lower than that of the wild-type protein but that are still able to fold. To select variants of CI2 that are more stable than the starting point we opted to use a destabilized background as starting point. The I57A variant of CI2 is significantly destabilized ($\Delta G_f = 14$ kJ mol$^{-1}$) and results in a high GFP signal in the sensor system (Fig. 1a). With a library of random mutations in a background of I57A there will be a larger dynamic range in the GFP signal towards more stable variants with less GFP and this library will be suited for selecting variants of CI2 with stabilities higher than I57A. Consequently, we prepared two libraries of random mutations with expected mutation frequencies of 0–4 amino acid residues per gene and sizes of 16,000–80,000 in the background of the wild-type sequence and of the I57A sequence, respectively.

The mutant libraries were expressed in the dual-sensor system in *E. coli* and analysed by FACS. To screen for destabilized protein variants in the library in the WT background, cells were sorted in three successive rounds for high GFP fluorescence, defined as the upper ~1 % of the GFP signal. In the same way, the library made in the I57A background was repeatedly sorted for the lower ~1% of the GFP signal. Both libraries were also selected for red fluorescence to ensure that the target protein is expressed and to minimize the fraction of clones with internal stop codons.

**Selecting variants for further analysis.** To identify the protein variants in the two final libraries we randomly selected 118 clones

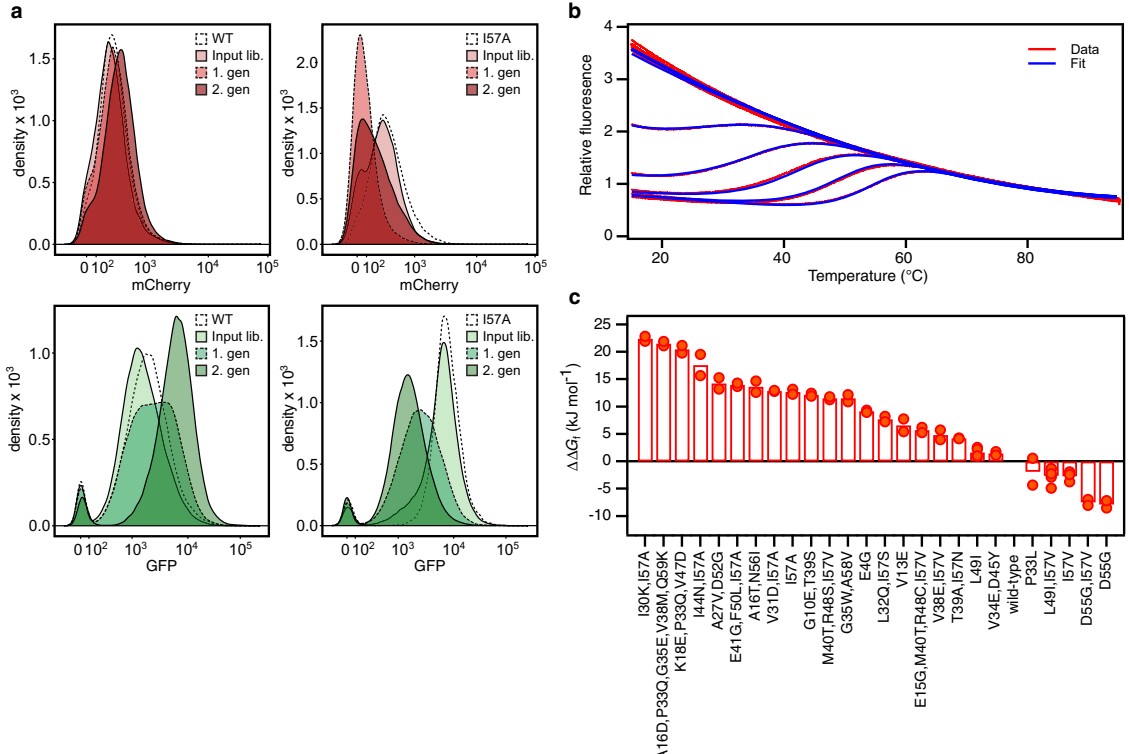

**Fig. 1 Selection and stability of CI2 variants. a** FACS profiles of *E. coli* cultures expressing libraries of random mutations in wild-type CI2 (left column) and in CI2,I57A (right column). mCherry fluorescence and GPF fluorescence are shown in the upper and lower rows, respectively. Profiles for the background variants of CI2, the input mutant library and the two rounds of sorting are shown. **b** Thermal unfolding curves of CI2,I57A measured by fluorescence at 350 nm at 13 concentrations of GuHCl ranging from 0 to 5 M. The experimental data are shown in red and the fits to a model for two-state folding as blue lines. **c** Difference in the free energy for folding, $\Delta\Delta G_f$, relative to wild-type CI2 for 26 variants. The $\Delta G_f$ for each variant was found by non-linear global fits of fluorescence unfolding data. For most variants two denaturant series were fitted ($n = 2$), except for L49I ($n = 4$), I57V ($n = 3$) and L49I/I57V ($n = 6$).

from the library starting from the wild-type sequence and 289 clones from the library starting from the I57A sequence by collection of single cells from the last rounds of FACS screening. These clones were characterised by Sanger sequencing. In addition, we also sequenced the final I57A library by next-generation sequencing (NGS). In total we found 71 unique sequences without stop codons, deletions or insertions in the region encoding the 64 amino acid residues of CI2. 41 of the sequences were from the wild-type library and 30 sequences were from the I57A library (Supplementary Fig. 1).

To express and purify the 71 CI2 variants, they were subcloned into pET11a without the hexa-His tag, which is part of the folding sensor system. The presence of this C-terminal His-tag interferes with key interactions of the CI2 C-terminal carboxylate and compromises the stability of the protein during purification. However, the destabilization gives just the right stability window for finding variants with altered stability in the sensor system. Thus, the FACS selection was done on CI2 libraries with the His-tag, whereas the in vitro stability measurements were performed on CI2 variants without the His-tag. Although we selected for protein variants that do not activate the misfolding sensor, some of the variants still did not behave well in the expression system and did not result in pure and stable protein or resulted in protein with multistate folding behaviour. We thus ended with 13 unique variants in the wild-type background and 12 unique variants in the I57A background that could be used for stability measurements. In the set of variants that we have analysed, we also included L49I and D55G (*vide infra*).

To measure the in vitro stability for folding we used our recently described combined two-dimensional thermal and chemical protein unfolding assay, where the unfolding of the protein is followed by the change in intrinsic Trp fluorescence as the temperature is increased at multiple concentrations of

denaturant (Fig. 1b)[34]. The stabilities of the 27 CI2 variants cover a broad range from $-7.4\,\mathrm{kJ\,mol^{-1}}$ to $-38.5\,\mathrm{kJ\,mol^{-1}}$ including five variants that are more stable than the wild-type protein (Table 1 and Fig. 1c). We find a strong correlation between $\Delta G_f$ at 25 °C and the melting temperature, $T_m$, which may thus be used as an additional parameter for comparing the stability. Except P33L, all variants selected in the wild-type background are destabilized and scattered throughout the sequence. P33L is stabilized by $1.5\,\mathrm{kJ\,mol^{-1}}$ but aggregates when the temperature is above 50 °C unless [GuHCl] > 2 M. All variants selected in the I57A background that are more stable than this background have a valine at position 57; we note that our starting point was designed to avoid random reversion to isoleucine by single nucleotide mutations (see Discussion). From previous work, it is known that I57V is slightly more stable than the wild-type protein[32], and valine at position 57 is also preferred in CI2 from many other plant species[35]. Most other mutations that occur together with I57V are less stable than I57V alone, but still more stable than the I57A background. There are however two exceptions. Both the L49I/I57V and D55G/I57V are even further stabilized than I57V. Both I at position 49 and G at position 55 are often seen in CI2 from other species[35]. The L49I and D55G variants have not previously been characterized so we also included these single variants in our analysis.

**Comparing with computational stability predictors**. To compare how well the stability of the selected set of CI2 variants can be predicted computationally we used FoldX, Rosetta and a sequence-based method that analyses variability in a multiple sequence alignment of homologous sequences (hereafter referred to as SEQ), to calculate $\Delta\Delta G_f$ for each variant (Fig. 2). The overall

**Table 1 Thermodynamic data for purified variants of CI2 identified with the folding sensor.**

| Variant | $T_m$ (K) | $\Delta H_m$ (kJ mol$^{-1}$) | $\Delta C_p$ (kJ mol$^{-1}$ K$^{-1}$) | $m$ (kJ mol$^{-1}$ M$^{-1}$) | $\Delta G_f$ (kJ mol$^{-1}$) | [D]$_{50\%}$ (M) | $\Delta\Delta G_f$ (kJ mol$^{-1}$) |
|---|---|---|---|---|---|---|---|
| [a]I30K, I57A | 327.4 ± 3.1 | −99 ± 11 | −0.4 ± 0.8 | 8.8 ± 1.8 | −8.2 ± 0.6 | 0.9 ± 0.2 | |
| A16D, P33Q, G35E, V38M, Q59K | 333.3 ± 1.9 | −134 ± 9 | −2.6 ± 0.3 | 5.4 ± 0.5 | −9.1 ± 0.6 | 1.7 ± 0.2 | |
| K18E, P33Q, V47D | 336.0 ± 1.7 | −147 ± 10 | −2.9 ± 0.1 | 4.3 ± 0.2 | −10.1 ± 1.0 | 2.3 ± 0.2 | |
| [a]I44N, I57A | 339.0 ± 2.4 | −163 ± 28 | −2.6 ± 0.4 | 7.4 ± 1.5 | −13.0 ± 2.8 | 1.8 ± 0.5 | |
| A27V, D52G | 341.7 ± 1.0 | −211 ± 19 | −3.6 ± 0.3 | 6.0 ± 0.8 | −16.4 ± 1.5 | 2.7 ± 0.4 | |
| [a]E41G, F50L, I57A | 331.6 ± 0.5 | −191 ± 3 | −1.5 ± 0.1 | 11.5 ± 0.2 | −16.6 ± 0.4 | 1.4 ± 0.0 | |
| A16T, N56I | 339.4 ± 0.6 | −227 ± 19 | −4.1 ± 0.3 | 7.0 ± 0.4 | −17.0 ± 1.5 | 2.4 ± 0.2 | |
| [a]V31D, I57A | 344.7 ± 0.5 | −222 ± 0 | −3.7 ± 0.0 | 7.0 ± 0.0 | −17.7 ± 0.1 | 2.5 ± 0.0 | |
| [a]I57A | 333.1 ± 0.5 | −269 ± 14 | −5.4 ± 0.5 | 11.6 ± 0.6 | −17.9 ± 0.6 | 1.5 ± 0.1 | |
| G10E, T39S | 339.7 ± 0.7 | −238 ± 16 | −4.0 ± 0.7 | 8.5 ± 0.5 | −18.4 ± 0.4 | 2.2 ± 0.1 | |
| [a]M40T, R48S, I57V | 344.3 ± 0.3 | −235 ± 9 | −3.8 ± 0.3 | 7.1 ± 0.1 | −19.1 ± 0.4 | 2.7 ± 0.1 | |
| G35W, A58V | 336.8 ± 0.1 | −281 ± 19 | −5.7 ± 0.5 | 6.9 ± 0.2 | −19.1 ± 0.9 | 2.8 ± 0.1 | |
| E4G | 341.5 ± 0.1 | −259 ± 7 | −4.0 ± 0.2 | 7.3 ± 0.1 | −21.4 ± 0.3 | 2.9 ± 0.1 | |
| L32Q, I57S | 348.1 ± 0.6 | −275 ± 5 | −4.4 ± 0.0 | 6.1 ± 0.2 | −22.9 ± 0.7 | 3.8 ± 0.2 | |
| V13E | 343.1 ± 0.2 | −280 ± 23 | −4.1 ± 0.5 | 9.6 ± 0.6 | −24.0 ± 1.6 | 2.5 ± 0.2 | |
| [a]E15G, M40T, R48C, I57V | 347.6 ± 0.1 | −281 ± 8 | −4.1 ± 0.1 | 8.3 ± 0.2 | −24.9 ± 0.7 | 3.0 ± 0.1 | |
| [a]V38E, I57V | 348.8 ± 0.3 | −295 ± 16 | −4.4 ± 0.3 | 8.9 ± 0.5 | −25.8 ± 1.2 | 2.9 ± 0.2 | |
| T39A, I57N | 343.5 ± 0.2 | −312 ± 0 | −4.7 ± 0.0 | 10.6 ± 0.1 | −26.4 ± 0.1 | 2.5 ± 0.0 | |
| L49I | 350.3 ± 0.2 | −322 ± 8 | −4.6 ± 0.1 | 8.7 ± 0.2 | −28.9 ± 0.8 | 3.3 ± 0.1 | 3.4 ± 0.1 |
| V34E, D45Y | 344.3 ± 0.0 | −363 ± 7 | −6.0 ± 0.2 | 8.4 ± 0.2 | −29.2 ± 0.5 | 3.5 ± 0.1 | |
| wild-type | 352.3 ± 0.2 | −322 ± 11 | −4.3 ± 0.2 | 8.2 ± 0.1 | −30.5 ± 0.8 | 3.7 ± 0.1 | 0 |
| P33L | 354.1 ± 0.5 | −330 ± 43 | −4.2 ± 0.7 | 8.3 ± 0.8 | −32.5 ± 3.5 | 3.9 ± 0.6 | |
| [a]L49I, I57V | 356.2 ± 0.3 | −332 ± 13 | −4.2 ± 0.2 | 7.9 ± 0.4 | −32.9 ± 1.3 | 4.2 ± 0.3 | −3.8 ± 0.1 |
| [a]I57V | 354.5 ± 0.2 | −349 ± 12 | −4.7 ± 0.2 | 8.3 ± 0.2 | −33.3 ± 0.9 | 4.0 ± 0.1 | −2.2 ± 0.1 |
| [a]D55G, I57V | 361.6 ± 0.3 | −364 ± 13 | −4.3 ± 0.2 | 8.5 ± 0.2 | −38.1 ± 0.8 | 4.5 ± 0.1 | −6.5 ± 0.2 |
| D55G | 361.1 ± 0.1 | −378 ± 10 | −4.7 ± 0.1 | 8.5 ± 0.2 | −38.4 ± 1.0 | 4.5 ± 0.2 | −6.9 ± 0.3 |

[a]Variant selected from the I57A library.

$T_m$, is the melting temperature in the absence of denaturant. $\Delta H_m$ is the enthalpy change for folding at $T_m$. $\Delta C_p$ is the change in heat capacity for folding. $m$ is the $m$-value for GuHCl unfolding. $\Delta G_f$ is the free energy for folding at 298 K. [D]$_{50\%}$ is the midpoint for the GuHCl unfolding at 298 K and $\Delta\Delta G_f$ is the change in stability at 298 K relative to wild-type from a fit of six variants with a common $m$-value of 8.4 ± 0.3 kJ mol$^{-1}$ M$^{-1}$. The errors are the propagated standard errors from the global fits of the data.

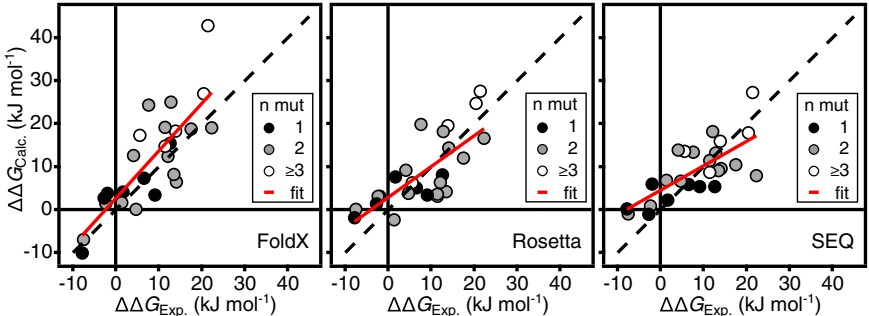

**Fig. 2 Correlations between experimentally determined and predicted $\Delta\Delta G_f$ values.** The predicted values were calculated by FoldX, Rosetta and SEQ as indicated in the lower right part of each panel. The dotted line shows the identity line and the solid red line is the best fit straight line. Each data point represents one of the variants in Table 1 and the data points are coloured according to the number of amino acid substitutions (black—one substitution; grey—two substitutions; white—three or more substitutions). The scale for the SEQ method is in arbitrary units.

performances of these computational methods on our CI2 variants are similar to benchmarks tests on other sets of proteins[24,25]. The Pearson's correlation coefficients ($r$) of the experimentally determined values with those calculated using FoldX, Rosetta and SEQ are 0.81, 0.74 and 0.72, respectively. In general, the three stability predictors agree in the overall effect of mutations, but they are inconsistent in the exact value. Importantly, while the methods are relatively good at predicting destabilizing effects, they are generally not able to predict stabilized variants, though FoldX does predict D55G to be highly stabilizing.

**Analysis of double mutant cycles.** In an attempt to understand better the origin of the increased stability of the two double mutants (D55G/I57V and L49I/I57V), we performed a more detailed analysis of the double mutation cycles from the wild-type through the single mutants to the double mutants. To compare the variants in the two double mutation cycles we re-analysed the stability data assuming a common $m$-value of 8.4 ± 0.3 kJ/mol/M

corresponding to the average of the $m$-values for wild-type, L49I, D55G, I57V, D55G/I57V and L49I/I57V (Table 1). As long as there is no significant change in the solvent accessible surface area exposed upon unfolding, the $m$-value is also expected not to change[36]. As done previously[32], we have therefore used the average $m$-value for comparing differences in the free energy for folding ($\Delta\Delta G_f$). As the $m$-value is not a fitting parameter in this analysis the standard errors of the other fitting parameters decreases and cannot be directly compared to results presented in Fig. 1c and Table 1. The normalized stability curves originating from this analysis are shown in Fig. 3a. Relative to the wild-type, I57V is stabilized by $\Delta\Delta G_f = −2.2 ± 0.1$ kJ mol$^{-1}$ (Fig. 3b). For D55G $\Delta\Delta G_f = −6.9 ± 0.3$ kJ mol$^{-1}$ and combining D55G and I57V leads to no further stabilization ($\Delta\Delta G_f = −6.5 ± 0.2$ kJ mol$^{-1}$). Indeed, the two mutations have an unfavourable synergistic effect, $\Delta\Delta\Delta G_f$, of 2.6 ± 0.4 kJ mol$^{-1}$. In contrast, introducing L49I, which on its own destabilizes by $\Delta\Delta G_f = 3.4 ± 0.1$ kJ mol$^{-1}$, together with I57V results in a total stabilization of the L49I/I57V double mutant of $\Delta\Delta G_f = −3.8 ± 0.1$ kJ mol$^{-1}$. In this case the synergistic effect of introducing both L49I and I57V is $\Delta\Delta\Delta G_f = −5.1 ± 0.2$ kJ mol$^{-1}$.

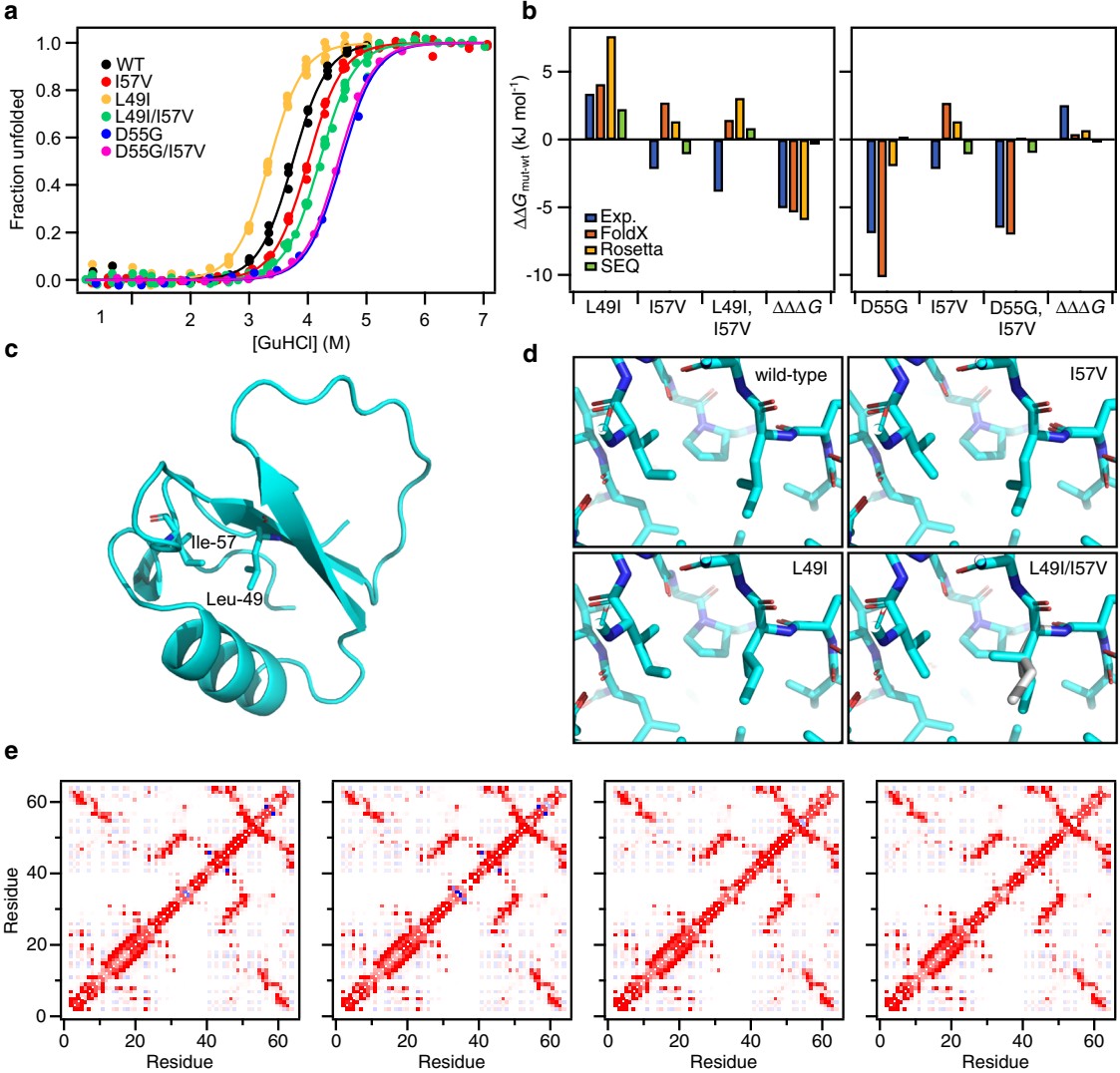

**Fig. 3 Stability and structural analysis of stabilized CI2 double mutants. a** Equilibrium stability curves of CI2 mutants at 25 °C. The experimental data (filled circles) were fit to a model for two state folding (solid line) keeping the $m$-value fixed at 8.4 kJ mol$^{-1}$ M$^{-1}$. The data are here normalized to show the degree of unfolding for direct comparison and visualization only. **b** Differences in stability relative to wild-type CI2, $\Delta\Delta G_f$, for the variants in the L49I/I57V (left) and D55G/I57V (right) double mutant cycles. Experimental $\Delta\Delta G_f$ as well as $\Delta\Delta G_f$ predicted by FoldX, Rosetta and SEQ are shown. The $\Delta\Delta\Delta G$ is the additional contribution to the conformational stability of the double mutants compared to the sum of the contribution of the single mutants, $\Delta\Delta G_{f,mut1/mut2} - (\Delta\Delta G_{f,mut1} + \Delta\Delta G_{f,mut2})$. The energy for the SEQ method is in arbitrary units. **c** Overview of the structure of CI2 with the locations of L49 and I57. **d** Structural details around positions 49 and 57 in the four CI2 variants in the wild-type to L49I/I57V double mutant cycle. **e** Residue wise contact maps. The pairwise interaction energies were calculated from AMBER FF99 using IEM[60]. The colour scale is from dark red ($-8$ kJ mol$^{-1}$) to dark blue ($8$ kJ mol$^{-1}$).

To evaluate if the observed effects of the double mutants could have been predicted, we compared with the expected $\Delta\Delta G_f$ from FoldX, Rosetta and SEQ. Particularly, FoldX does a good job in predicting the effects of the L49I and D55G mutations (Fig. 3b). None of the tools work well for predicting $\Delta\Delta G_f$ for I57V or L49I/ I57V. FoldX predicts the effect of D55G/I57V rather well, but this can be attributed to the dominating effect of the D55G mutation that was well predicted on its own. It appears as if $\Delta\Delta\Delta G_f$ of L49I/ I57V is well predicted by both FoldX and Rosetta (Fig. 3b). However, the individual $\Delta\Delta G_f$ used for the calculations are not correct and in most cases the signs of the values are incorrect. We thus conclude that the outcome of the double mutations could not have been predicted reliably.

**Structural analysis**. The positive synergistic effect of L49I and I57V is interesting to examine in more detail to gain insight into how proteins may be designed or evolve to become more stable.

We, therefore, determined the crystal structures of all four CI2 variants in the wild-type to L49I/I57V double mutant cycle (Table 2). Overall, the structures of L49I, I57V and L49I/I57V are highly similar to the structure of the wild-type (Fig. 3c) with RMSDs for the backbone of 0.16, 0.45 and 0.17 Å. The larger RMSD for I57V is a result of the overhand loop in this structure adopting an alternative conformation compared to the other three structures. This conformation is similar to the conformation of the overhand loop in the older crystal structure of wild-type CI2 (PDB-code 2CI2)[37,38]. Excluding residues 44–50 in this loop from the comparison reduces the backbone RMSD to the wild-type to 0.14 Å, 0.16 Å and 0.15 Å for L49I, I57V and L49I/I57V, respectively.

Around the mutated residues the structural changes are minimal (Fig. 3d). The Val at position 57 in I57V and L49I/ I57V superimpose, except for the missing δ1 methyl group, with the Ile at position 57 in the wild-type and L49I. The Ile at position

**Table 2 Crystallographic statistics.**

| Data set | CI2 WT | CI2 I57V | CI2 L49I | CI2 L49I,I57V |
|---|---|---|---|---|
| PDB code | 7A1H | 7A3M | 7AOK | 7AON |
| *Data collection* | | | | |
| Beamline | N.A. | DESY, B13 | DESY, B13 | DESY, B13 |
| Wavelength (Å) | 1.5406 | 0.97625 | 0.97625 | 0.97625 |
| Temperature (K) | 150 | 100 | 100 | 100 |
| Space group | P622 | P622 | P622 | P622 |
| a, b, c (Å) | 68.27, 68.27, 52.60 | 68.44, 68.44, 52.10 | 68.47, 68.47, 52.94 | 68.39, 68.39, 52.66 |
| $\alpha,\beta,\gamma$ (°) | 90, 90, 120 | 90, 90, 120 | 90, 90, 120 | 90, 90, 120 |
| Protein molecules in asymmetric unit | 1 | 1 | 1 | 1 |
| Resolution range (Å) | 15.0–1.90 | 59.3–1.01 | 59.3–1.87 | 59.2–1.26 |
| Number of reflections collected | 52781 | 968735 | 183674 | 571649 |
| Number of unique reflections | 6063 | 29927 | 5666 | 15866 |
| Multiplicity | 8.7 (5.2) | 32.4 (5.9) | 32.3 (36.4) | 36.0 (34.1) |
| Completeness (%) | 99.7 (99.7) | 95.4 (60.7) | 88.0 (100.0) | 94.7 (63.4) |
| $R_{pim}$ | 0.033 (0.332) | 0.015 (0.506) | 0.019 (0.325) | 0.016 (0.462) |
| $CC_{1/2}$ | 0.999 (0.565) | 0.9995 (0.4978) | 0.998 (0.867) | 0.9995 (0.7034) |
| $\langle I/\sigma(I)\rangle$ | 10.4 (1.7) | 22.9 (1.2) | 22.4 (2.5) | 22.6 (1.5) |
| *Refinement statistics* | | | | |
| $R_{work}/R_{free}$ (%) | 20.5/25.2 | 14.9/18.1 | 22.0/24.6 | 16.2/21.9 |
| Number of atoms | 551 | 625 | 538 | 598 |
| Water molecules | 28 | 98 | 18 | 60 |
| $\langle B\rangle$ (Å$^2$) | 23.4 | 14.9 | 37.2 | 24.2 |
| Minimal estimated coordinate errors (Å) | 0.123 | 0.003 | 0.015 | 0.005 |
| *RMS deviations from ideal geometry* | | | | |
| Bond lengths (Å) | 0.020 | 0.031 | 0.020 | 0.030 |
| Bond angles (°) | 2.078 | 2.516 | 2.065 | 3.143 |
| Chiral volume (Å$^3$) | 0.111 | 0.140 | 0.109 | 0.192 |
| *Ramachandran plot* | | | | |
| Preferred regions (%) | 98.4 | 98.3 | 96.8 | 96.7 |
| Allowed regions (%) | 1.6 | 1.7 | 3.2 | 3.3 |
| Outliers (%) | 0 | 0 | 0 | 0 |

Numbers in parenthesis refer to outer shell data.

49 in L49I is oriented similarly to the Leu in wild-type and I57V. In the structure of the double mutant, however, we observe that the Ile at position 49 is found in two alternative conformations (Supplementary Fig. 2). The minor conformation, accounting for roughly 30% of the electron density, has a conformation similar to that seen in L49I. In the major conformation that accounts of the remaining 70% of the electron density, the γ2 methyl is rotated approximately 120° and points towards the position where the δ1 methyl group of the Ile at position 57 would be in the wild-type structure.

Analysis of the pairwise interaction energies at the residue level (Fig. 3e) suggests that much of the stabilizing effect of the I57V mutation originates from an unfavourable interaction between I57 and Q59 that is observed in structures of both the wild-type and L49I. The effect of the L49I mutation, which destabilizes the wild-type, but stabilizes the I57V variant is more subtle and not easily explained from the crystal structures. It appears that some of the destabilization of the L49I mutation is a long-range effect resulting in several less favourable interactions among residues 57-62. These negative effects are relieved in the double mutant (Fig. 3e).

## Discussion

Using our bacterial stability sensor, we selected 25 stable and cooperatively folded variants of CI2 from libraries of random mutations. We thus demonstrate that the system can be used as a screening assay for methods like directed evolution and deep mutational scanning to select protein variants with both stabilizing and destabilizing effects originating from mutant libraries. In one of the libraries, we introduced the destabilizing I57A

mutation to improve the dynamical range of our experiments. The Ala was introduced by changing the 57th codon to GCG. To mutate this codon into an Ile codon three base substitutions are needed. It is thus highly unlikely that a revertant to the wild-type will be generated by error prone PCR, and we indeed did not observe the wild-type sequence in this library. Instead, the most abundant mutant after selection was I57V which can be made by a single base substitution from the Ala codon. As I57V is more stable than the wild-type this demonstrates the efficiency of the selection system. It also demonstrates that our settings in the error prone PCR did not result in a very broad library, and indeed most codons in the selected variants are just one base substitution from the wild-type sequence. The large gain in stability of the I57V variant over the I57A background is a challenge to the folding sensor as the GFP signal reaches close to its minimal level with just one base substitution. We anticipate that for a naturally evolved protein of low stability several mutations, which each provide a small change in the stability and thus in the GFP signal, would accumulate through multiple rounds of mutagenesis and selection and eventually lead to a more stable protein.

As all the variants selected from the I57A background carry either the I57A or the I57V it is not surprising that many double mutants were selected. Also, all variants selected in the I57A background that are more stable than I57A carry I57V, suggesting that not many single variants in the library stabilize CI2. The latter is supported by in silico saturation mutagenesis by Rosetta and FoldX that only predict very few stabilizing single point mutations and only 6 and 3, respectively, to be stabilized by more than 2 kJ mol$^{-1}$ (Supplementary Fig. 3). Two of the double

mutants we found (L49I/I57V and D55G/I57V) were, however, highly stabilized, both relative to the I57A background but also more than the wild-type protein. In an attempt to understand the origin of this increased stability we also included the single mutants L49I and D55G in our analysis to make a thermodynamic double mutant cycle. For D55G/I57V the increased stability is almost completely an effect of the D55G mutation. We suggest that it is a result of the residue at position 55 being located in the $\alpha_L$ region of the Ramachandran map, where Gly is even more common than Asp[39]. Repulsive interactions with nearby E14 could also play a role. For L49I/I57V on the other hand there is a large non-additive effect. From the crystal structures a few subtle changes in the interaction energies that could contribute to stabilization were identified. L49 and I57 are both part of the folding nucleus in CI2 and very important for the stability of the folded state of CI2[32]. Previous stability studies of Ala mutants showed that there is strain in the native CI2 structure between I57 and A16 but not between L49 and A16[40]. The interaction between L49 and I57 was not analysed in that work. Computational analysis by Hilser et al suggested that mutation of L49 would propagate energetically to the rest of the protein[41]. Together with our results these observations by others suggest that the packing of hydrophobic core around L49 and I57 in wild-type CI2 is suboptimal, which is alleviated by subtle changes from combining L49I and I57V mutations.

The prediction methods that we used were not able to pick up the effects of the I57V and L49I mutations that we observe. One explanation for this is that the changes FoldX and Rosetta make to a structure to accommodate a mutation are local. If long range changes are important to explain the change in stability the methods will miss them. Furthermore, errors may accumulate when multiple mutations are introduced (Fig. 2).

In conclusion, we have generated a set of variants in CI2 with varying stabilities and most of them containing multiple amino acid substitutions. Our selection procedure has demonstrated its strength as we have identified double mutants that stabilize the protein. Such mutants are presumably rare in CI2. Of particular interest is the synergistic effect of the two substitutions in L49I/I57V. Although we see small structural changes in the structure compared with the other structures in the mutant cycle, the presence of two distinct conformations of the Ile at position 49 in the double mutant could point to effects of conformational entropy or conformational changes also contributing.

## Methods

**CI2 mutant libraries**. The wild-type CI2 sequence used here is UniProt: P01053, residues 22–84 with an additional N-terminal Met. The numbering we use, start at this Met, which is the numbering system commonly used in the literature. cDNA encoding this sequence was inserted into a pET22-mCherry vector between the NdeI and HindIII restriction sites to preserve the translation coupling using Gibson assembly. The QuickChange Lightning II mutagenesis kit (Agilent) was used to create the I57A variant in the CI2_WT_pET22_mCherry vector. Mutant libraries of CI2 WT and CI2 I57A were generated using the GeneMorph II Random mutagenesis kit (Agilent). The mutation frequency was aimed at 0-4 amino acid substitutions per gene by adjusting the initial target DNA and the number of amplification cycles. The PCR product from the random mutagenesis was used as a MEGAprimer for the insertion of the mutants into the CI2 WT or CI2 I57A backgrounds. The PCR products were transformed into MegaX DH10B T1$^R$ Electrocomp Cells (Invitrogen) following the manufacturer's instructions and everything was plated on LB agar plates with 100 μg mL$^{-1}$ ampicillin.

The CI2 libraries were transformed into electrocompetent Rosetta2(DE3)pLysS pSEVA631-IBpAP-GFP-ASV cells[5]. After recovery, the transformants were directly inoculated in 3 mL LB medium containing 100 μg mL$^{-1}$ ampicillin, 25 μg mL$^{-1}$ chloramphenicol and 50 μg mL$^{-1}$ spectinomycin and grown overnight at 37 °C and 180 rpm. Cells were transferred into fresh medium and grown at 30 °C and 250 rpm to an OD$_{600}$ of 0.5–0.7. Expression was induced by addition of 0.5 mM IPTG and the growth temperature of the culture was shifted to 30 °C.

One hour after induction cells were analysed by flow cytometry (Instrument: BD FACS-Aria SORP cell sorter; Laser 1: 488 nm: >50 mW, Filter: 505LP, 515/20-nm FITC; Laser 2: 561 nm: >50 mW; Filter: 600LP, 610/20-nm PE-Texas Red).

100,000 cells expressing a CI2 WT mutant protein with increased GFP signal, and 100,000 cells expressing a CI2 I57A mutant protein with decreased GFP signal were sorted (gating strategy in Supplementary Fig. 4) in 2 mL LB medium supplemented with antibiotics and grown overnight at 37 °C and 300 rpm. To further enrich the *E. coli* fraction harboring proteins with altered protein stability, protein expression was induced again and cells (100,000 events) were sorted as described above. The following day, the sorted cell population was analysed 1 h after induction of protein expression by flow cytometry (Instrument: BD FACS-Aria SORP cell sorter; Laser 1: 488 nm: >50 mW, Filter: 505LP, 515/20-nm FITC; Laser 2: 561 nm: >50 mW; Filter: 600LP, 610/20-nm PE-Texas Red). Single cells were sorted directly into 100 μl LB supplemented with antibiotics in 96 well culture plates and grown overnight at 37 °C and 300 rpm. The single cells were characterized by Sanger sequencing.

**Library preparation for NGS**. Samples from the CI2 WT library were extracted from each step of FACS selection for NGS. 2 mL culture was centrifuged for 10,000 × g for 2 min and plasmids purified using the NucleoSpin Plasmid kit (Machery-Nagel). The CI2 genes were amplified using the Phusion Hot Start II DNA Polymerase (Thermo Scientific) and region of interest-specific primers with overhang adapters. The PCR amplicons were purified using AMPure XP beads (Beckman Coulter) following the manufacturer's instructions. Dual indices and Illumina sequencing adapters (Nexteras XT Index kit (Nextera v2 D)) were attached using the KAPA HiFi HotStart DNA polymerase. The amplicon libraries were purified using AMPure XP beads (Beckman Coulter) following the manufacturer's instructions. Concentrations of the purified amplicon libraries were quantified using a Qubit 2.0 Fluorometer and the dsDNA broad range kit. To determine the average bp length, a bioanalyzer and the DNA 1000 kit (Agilent) was used following the manufacturer's instructions. The libraries were normalized to 10 nM and all libraries were pooled. The samples were spiked with 5% Phi-X control DNA (Illumina) and loaded onto the flow cell and sequenced on and then applied onto an Illumina MiSeq instrument.

**Cloning, expression and stability of single sorted CI2 clones**. Glycerol stocks of single cells sorted from the CI2 libraries were used as DNA template in colony PCR using the Phusion Hot Start II High-Fidelity DNA polymerase (Thermo Scientific). CI2 genes were amplified and ligated into a pET11a vector using the NdeI and BamHI restriction sites. All clones were verified using Sanger sequencing. All protein expression were performed in BL21(DE3)pLysS. For small scale protein expression, the bacteria were grown in 2 ml ZYM5052 autoinduction media in a 24 well cell culture plate with incubation at 37 °C for 4 h followed by 20 h at 20 °C. Low expression level plasmids were expressed in 50 mL ZYM5052 autoinduction media. Cells were harvested using centrifugation at 5000 × g for 15 min. The pellet frozen at −20 °C and resuspended in 10 mM Na-acetate, pH 4.4 before centrifugation at 20,000 × g for 30 min. The supernatant was further diluted in the same buffer. The samples were applied onto a 1 ml Resource S column equilibrated with 20 mM Na-acetate, pH 4.4 and step eluted with 20 mM Na-acetate, pH 4.4, 1 M NaCl. The peak fraction was applied onto a Superdex 75 16/85 column equilibrated with 50 mM NH$_4$HCO$_3$, and the fractions containing CI2 were collected and lyophilized before dissolving, in 50 mM MES, pH 6.25. For expression of protein for structure determination the bacteria were grown in LB media. The expression was induced by 0.4 mM IPTG at OD$_{600}$ 0.6–0.8. Cells were harvested by centrifugation at 5000 × g for 15 min and resuspended in 25 mM Tris-HCl, pH 8, 1 mM EDTA before lysis by two freeze-thaw cycles. The sample was cleared by centrifugation at 20,000 × g at 4 °C. Polyethylenimine was added to a concentration of 1% and the sample centrifuged for 15 min at 20,000×g. Ammonium sulphate was added to the supernatant to 70% saturation, and left for 30 min at 4 °C before centrifugation at 20,000 × g. The pellet was resuspended in 25 mM Tris-HCl, pH 8 and heated at 40 °C until all precipitate was solubilized. The samples were centrifuged at 20,000 × g for 10 min before size exclusion chromatography in 10 mM NH$_4$HCO$_3$. Peak fractions were pooled and lyophilized and finally resuspended in MilliQ water and dialysed against water. We note that while we attempted to express 71 variants of CI2, a number of them displayed did not yield soluble protein or give useful data in the equilibrium unfolding experiments. We hypothesise that this could be caused by a subset of the variants bypassing the folding sensor by weakening the interaction with DnaK as suggested by predicting DnaK binding site with the Limbo algorithm[42] (Supplementary Fig. 5).

**Equilibrium unfolding**. Protein concentrations were determined by absorbance at 280 nm measured on a NanoDrop 1000 due to the low volume samples. Equilibrium stability in GuHCl was measured with a final protein concentration of 10 μM at 6–16 concentrations of GuHCl evenly distributed in the range from 0 to 5, 6 or 7 M depending on the stability of the variant[34]. The degree of unfolding was followed by fluorescence measurements on a Prometheus NT.48 (NanoTemper) using Prometheus NT.48 high sensitivity capillaries. The temperature was ramped from 15 to 95 °C with a temperature increment of 1 °C min$^{-1}$. Global analysis of temperature and solvent denaturation was performed as described[34].

**Computational prediction of stability and DnaK binding**. Version 4 of the FoldX energy function[19] was used to estimate the free-energy change upon mutations of

CI2, using the coordinates of the PDB entry 2CI2[37]. The RepairPDB function of FoldX was first applied to the wild type structure. The resulting structure was used as input to the BuildModel function to generate the models of the investigated mutants and to evaluate their $\Delta\Delta G_f$.

The Rosetta energy function[15] in its cartesian version[43] was also used to estimate $\Delta\Delta G_f$, using the coordinates of PDB entry 2CI2. The wild-type structure was first relaxed in cartesian space with restrained backbone and sidechain coordinates. The resulting coordinates were then used to build the model of the investigated mutants and to evaluate their $\Delta\Delta G_f$ by means of the Cartesian_ddg function. The calculations were repeated on five independent runs, whose results were then averaged to obtain the final values reported in the manuscript. The resulting difference in stability was multiplied by 1.44 to bring the $\Delta\Delta G$ values from Rosetta energy units onto a scale corresponding to kJ mol$^{-1}$ (ref. [44]).

Looking at the mutational pattern observed in a multiple sequence alignment of homologous sequence, it is possible to build a global statistical model of the relative protein family variability[45], which takes into account not only single-site conservation, but also correlated mutations between site pairs. This approach aims at exploiting the structural and functional constraints encoded in the family evolution[46], assigning to each specific sequence a score related to the probability of being a good representative of that family[47]. Even if this measure is more related to the general fitness of the sequence, it can also be used *bona fide* to judge the effect of a specific mutation on protein stability[47,48]. To have a variation model which is statistically significant, we obtained a larger multiple sequence alignment containing CI2 homologues by building a hidden Markov model of the protein family, based on 4 iterations of Jackhmmer[49] and extracting the sequences from the Uniprot Uniref100 database[50]. Sequences containing more than 50% of gaps with respect to the wild-type sequence were excluded, together with the sequences sharing more than 90% of sequence identity, resulting in an alignment of 1198 sequences, corresponding to 942 independent sequences at the 95% identity level. We then used the asymmetric plmDCA algorithm[51,52] to calculate the parameters of the sequence model. The score of the wild-type sequence was then subtracted to the one of each analysed sequence to obtain the final values reported in the manuscript. The scripts and sequence alignment used in these analyses are available from https://github.com/KULL-Centre/papers/tree/master/2021/CI2-Hamborg-et-al.

**Crystallization, diffraction experiments and structure calculations**. CI2 WT, L49I and L49I/I57V were crystalized at 293 K in 40% $(NH_4)_2SO_4$, 50 mM Tris-HCl, pH 8.0 at a protein concentration of 75 mg ml$^{-1}$. CI2 I57V were crystalized in 0.1 M Tris-HCl, 8% PEG 8000, pH 8.5. Data for WT were collected on an inhouse setup with an Agilent SuperNova diffraction source (1.5406 Å) and an Atlas CCD detector. Data for L49I, I57V and L49I/I57V were collected at DESY, Hamburg, beamline P13 (0.9763 Å) equipped with a Pilatus 6M-F, S/N 60-0117-F detector.

The reflections were collected using autoPROC[53], which also scales and merges the data using the CCP4 programs Pointless and Aimless[54] as well as Staraniso (http://staraniso.globalphasing.org/cgi-bin/staraniso.cgi). The latter program was employed because of pronounced anisotropic distribution of reflections. The structures were solved by molecular replacement using the CCP4 program Phaser[55]. The initial search model was wild-type CI2 (PDB code 2CI2)[37]. Later molecular replacement solutions were obtained using the higher resolution structures described in the present paper. The structures were carefully examined, adjusted and refined with Coot and Refmac5, respectively[56,57]. To make sure that the structures were not in a domain swapped configuration[58], molecular replacement solutions were also sought using the domain swapped structure with PDB accession code 6QIZ. These consistently yielded significantly worse statistics.

**Statistics and reproducibility**. Each stability titration curve was made from at least 6 samples at different denaturant concentrations. We previously performed a systematic analysis of the minimal number of samples required for a robust analysis and found that five samples were sufficient for getting reliable results[34]. For most variants two denaturant series were fitted ($n = 2$), except for L49I ($n = 4$), I57V ($n = 3$) and L49I/I57V ($n = 6$). The reproducibility of the fitted $\Delta G_f$ at 25 °C is estimated to 1.2 kJ mol$^{-1}$ from the pooled standard deviation[59]. No additional statistical analysis was used in this work.

**Reporting summary**. Further information on research design is available in the Nature Research Reporting Summary linked to this article.

## Data availability

The atomic coordinates and structure factors for the structures presented in this work are deposited at the Protein Data Bank (https://www.wwPDB.org) under accession numbers 7A1H, 7A3M, 7AOK and 7AON. Source data for Fig. 1a-c Fig. 2a Fig. 3a and b are available as Supplementary Data 1–6. Other data included in the figures and that support the findings of this study are available from the corresponding author upon reasonable request.

## Code availability

The scripts and examples for the sequence-based stability analysis are available from https://github.com/KULL-Centre/papers/tree/master/2021/CI2-Hamborg-et-al.

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

## Acknowledgements

This work was supported by the Novo Nordisk Foundation [grant numbers NNF15OC0016360, NNF18OC0033926 and NNF20CC0035580]. The authors thank Pia Skovgaard for technical assistance. K.L.L. and K.T. are members of Integrative Structural Biology at the University of Copenhagen (www.isbuc.ku.dk). We thank Profs. F. Rousseau and J. Schymkowitz (VIB) for sharing the Limbo Software. We acknowledge excellent support at the P14 beamline operated by EMBL Hamburg at the PETRA III storage ring (DESY, Hamburg). We are grateful for support from the DANSCATT program of the Danish Council for Research and Innovation. We thank Thomas Lykke-Møller Sørensen (Aarhus University) for help with the data acquisition at DESY. Finally, we thank Pernille Harris for access to the in-house diffractometer at the Technical University of Denmark.

## Author contributions

A.T.N., K.L.L. and K.T. designed research; L.H., D.G., J.G.O. and JVR performed research; L.H., D.G., J.G.O., L.E.P., A.T.N., K.L.L. and K.T. analysed data; and L.H., K.L.L. and K.T. wrote the paper with inputs from all authors.

## Competing interests

A.T.N. is an inventor on a patent that covers the folding sensor system used in the current work. All other authors declare no competing interests.
