## [Peer Review File · Communications Biology]

Reviewers' Comments:

Reviewer #1:

Remarks to the Author:

In this study the authors have used a cell-based screen to look for mutants of CI2 which are either destabilized or stabilized, the motivation being to find a larger library of stabilized mutants which might act as a stringent test of methods for stability prediction (still a hard problem in protein science). Some novel stabilizing pairs of mutations are found and further studied in order to understand the subtle cooperative and anticooperative nature of the mutation pairs.

Overall, this is a high quality and thorough piece of work. I suggest the authors address the following:

1. To make the stability readout sufficiently sensitive that differences in stability could be detected, the authors started with a destabilized variant of CI2, I57A, which is sufficient to cause a response of the chaperone machinery that triggers the GFP fluorescence in their assay for stability. In the stabilized variants, one would have expected some mutations to this site as an obvious place to compensate for lost stability. However, it turns out that every one of the stabilized variants include such a mutation, I57V. This suggests that the only way to get a sufficient gain in stability to get a readout with the assay (and allowing only a few mutations) is to change position 57, so what one sees are I57V +/- some additional mutations that tag along for the ride. Is there any way this assay may be tweaked in future studies so one could look for more generally stabilizing mutations? e.g. looking at a slightly less destabilized variant or at a different protein?

Minor issues:

2. Figure 1 was very blurry and hard to read.
3. Fig 1 legend - "conformational stability" - native state stability?
4. Table 1 - Indicate temperature at which delta G is reported in caption (298 K).
5. Table 1 - For completeness, define the terms T_m , ΔH_m , ΔG_f etc. The sign convention used for ΔG_f seems opposite to the other quantities, which may be confusing (it was for me, anyway).

Reviewer #2:

Remarks to the Author:

This was an interesting study - it's essentially a practical demonstration of a method recently developed by the authors for high-throughput screening of protein variants for changes in stability.

I have one major issue with the design of the study, which is the decision to search for stabilizing variants only using the I57A background. The logic for this seems sound - the wild-type protein is so stable that it barely gives any GFP signal, but then the authors find that every single stabilizing variant they identified was observed in combination with another mutation at the same residue, I57V (which was already known to be more stable than wild type). Thus it seems to me that there was no point in making the I57A mutation in the first place, as everything they found included a (pseudo) reversion. Did the authors try selecting for stabilizing variants from a wild-type background? It seems this would have been much more efficient, and it's also hard for me to imagine that they didn't at least try this -

so why didn't it work?

It's also slightly disappointing to me that the authors found only 2 new stabilizing mutants. My intuition is that there are probably many more than this, and therefore this high-throughput method was not very effective at discovering them. However, I accept I could be wrong here, and maybe there really are very few stabilizing mutants to be discovered.

Finally, I think the "SEQ" method of predicting mutation stability needs to either be described in much more detail, or removed from the study. It sounds very interesting, so I hope there's a paper coming. It's hard for me to understand how it can distinguish between stabilizing and destabilizing mutations, unless it's assuming that stabilizing mutations are always more fit.

Despite these issues, I still think it is a very nice study and worthy of publication. The detailed analysis and mechanistic interpretation of the mutations including structural characterization is very good.

Rebuttal - Manuscript COMMSBIO-21-1085-T

Reviewers' comments:

Reviewer #1 (Remarks to the Author):

In this study the authors have used a cell-based screen to look for mutants of CI2 which are either destabilized or stabilized, the motivation being to find a larger library of stabilized mutants which might act as a stringent test of methods for stability prediction (still a hard problem in protein science). Some novel stabilizing pairs of mutations are found and further studied in order to understand the subtle cooperative and anticooperative nature of the mutation pairs.

Overall, this is a high quality and thorough piece of work.

We thank the reviewer for the nice comment.

I suggest the authors address the following:

1. To make the stability readout sufficiently sensitive that differences in stability could be detected, the authors started with a destabilized variant of CI2, I57A, which is sufficient to cause a response of the chaperone machinery that triggers the GFP fluorescence in their assay for stability. In the stabilized variants, one would have expected some mutations to this site as an obvious place to compensate for lost stability. However, it turns out that every one of the stabilized variants include such a mutation, I57V. This suggests that the only way to get a sufficient gain in stability to get a readout with the assay (and allowing only a few mutations) is to change position 57, so what one sees are I57V +/- some additional mutations that tag along for the ride. Is there any way this assay may be tweaked in future studies so one could look for more generally stabilizing mutations? e.g. looking at a slightly less destabilized variant or at a different protein?

The reviewer is correct that the Val at position 57 is dominating. This clearly shows that the selection method is efficient, but of course also reduces the likelihood of other stabilizing mutations or double mutants being selected. This will inevitably be a challenge if multiple selections are performed, and one very stable variant is obtained by a single base substitution. The situation would be very different for a less stable protein where several mutations accumulated though several rounds of mutagenesis and selection would eventually lead to a more stable protein. Furthermore, as pointed out in the response to reviewer 2 not many single point mutations are expected to stabilize CI2. To address the thoughts of the reviewer we have added the following to the ...

Minor issues:

2. Figure 1 was very blurry and hard to read.

We apologize for the bad quality of the figure, which was definitely not indented. We have now uploaded all figures as individual high quality pdf-files. In addition, we have changed the layout of the panels in Figure 1, but have not changed any of the contents of each panel.

3. Fig 1 legend - "conformational stability" - native state stability?

We have changed "Difference in the conformational stability,..." to "Difference in the free energy for folding,..."

4. Table 1 - Indicate temperature at which delta G is reported in caption (298 K).

See next point

5. Table 1 - For completeness, define the terms T_m , ΔH_m , ΔG_f etc. The sign convention used for ΔG_f seems opposite to the other quantities, which may be confusing (it was for me, anyway).

We have now defined all the parameters in the caption of the table. We thank the reviewer for catching that the signs were not consistent. They have now been corrected so they all correspond to the folding reaction as stated in the caption.

Reviewer #2 (Remarks to the Author):

This was an interesting study - it's essentially a practical demonstration of a method recently developed by the authors for high-throughput screening of protein variants for changes in stability.

I have one major issue with the design of the study, which is the decision to search for stabilizing variants only using the I57A background. The logic for this seems sound - the wild-type protein is so stable that it barely gives any GFP signal, but then the authors find that every single stabilizing variant they identified was observed in combination with another mutation at the same residue, I57V (which was already known to be more stable than wild type). Thus it seems to me that there was no point in making the I57A mutation in the first place, as everything they found included a (pseudo) reversion. Did the authors try selecting for stabilizing variants from a wild-type background? It seems this would have been much more efficient, and it's also hard for me to imagine that they didn't at least try this - so why didn't it work?

The reviewer is correct that it would have been much more straightforward to select directly from the wild-type background. However, wild-type CI2 results in a very low GFP signal in our folding sensor. This signal is only marginally different from the GFP signal from R48I (a previously known highly stabilized CI2 variant) which we tested during the development of the folding sensor. This analysis is presented in another manuscript which is currently under consideration for publication in another journal. A preprint of this manuscript that we also refer to in the current manuscript is available ([doi:10.1101/2020.09.18.303453](https://doi.org/10.1101/2020.09.18.303453)).

To emphasize the challenge in selecting stabilized variants directly from wild-type CI2 we have added the following sentences to the first paragraph of the results section:

During the development of the folding sensor, we compared the GFP signal from wild-type CI2 with that from the highly stabilized variant R48I and found only a marginal difference⁵. It will thus not be possible to separate stabilized variants from the wild-type or from variants with stabilities close to the that of the wild-type.

It's also slightly disappointing to me that the authors found only 2 new stabilizing mutants. My intuition is that there are probably many more than this, and therefore this high-throughput method was not very effective at discovering them. However, I accept I could be wrong here, and maybe there really are very few stabilizing mutants to be discovered.

We agree with the reviewer that it would have been great to find more stabilized variants. However, wild-type CI2 is already relative stable. To get a rough estimate of how many stabilizing single point mutations that may be found in CI2 we performed *in silico* single point saturation mutagenesis by Rosetta and FoldX and found only 6 and 3 point mutations that would stabilize CI2 by more than 2

kJ/mol. Six mutations are at position 48 and 55. The final three suggested by Rosetta only, are at position 1, which is dynamic.

We have added the following to the second paragraph of the discussion together with a new supplementary figure S3 that summarizes the *in silico* mutagenesis by Rosetta and FoldX.

...suggesting that not many single variants in the library stabilize CI2. The latter is supported by in silico saturation mutagenesis by Rosetta and FoldX that only predict very few stabilizing single point mutations and only 6 and 3 to be stabilized by more than 2 kJ mol⁻¹, respectively (Supplementary Figure 3).

We acknowledge that there potentially will be more double mutations that stabilize CI2, but that the size of our library may not cover enough sequences. To this end we already write the following in the first paragraph of the discussion:

It also demonstrates that our settings in the error prone PCR did not result in a very broad library, and indeed most codons in the selected variants are just one base substitution from the wild-type sequence.

Finally, I think the "SEQ" method of predicting mutation stability needs to either be described in much more detail, or removed from the study. It sounds very interesting, so I hope there's a paper coming. It's hard for me to understand how it can distinguish between stabilizing and destabilizing mutations, unless it's assuming that stabilizing mutations are always more fit.

We are happy that the reviewer finds our sequence based stability method of interest. It is based on already published ideas and we have now added a few more references to the text and expanded the explanation of how the alignment is performed. In addition, the script used for the analysis and the sequence alignment are now available via github. The text explaining the procedure now reads: *Looking at the mutational pattern observed in a multiple sequence alignment of homologous sequence, it is possible to build a global statistical model of the relative protein family variability⁴⁴, which takes into account not only single-site conservation, but also correlated mutations between site pairs. This approach aims at exploiting the structural and functional constraints encoded in the family evolution⁴⁵, assigning to each specific sequence a score related to the probability of being a good representative of that family⁴⁶. Even if this measure is more related to the general fitness of the sequence, it can also be used bona fide to judge the effect of a specific mutation on protein stability^{46,47}. To have a variation model which is statistically significant, we obtained a larger multiple sequence alignment containing CI2 homologues by building a hidden Markov model of the protein family, based on 4 iterations of Jackhmm⁴⁸ and extracting the sequences from the Uniprot Uniref100 database⁴⁹. Sequences containing more than 50% of gaps with respect to the wild-type sequence were excluded, together with the sequences sharing more than 90% of sequence identity, resulting in an alignment of 1198 sequences, corresponding to 942 independent sequences at the 95% identity level. We then used the asymmetric plmDCA algorithm^{50,51} to calculate the parameters of the sequence model. The score of the wild-type sequence was then subtracted to the one of each analysed sequence to obtain the final values reported in the manuscript. The scripts and sequence alignment used in these analyses are available from <https://github.com/KULL-Centre/papers/tree/master/2021/CI2-Hamborg-et-al>.*

Despite these issues, I still think it is a very nice study and worthy of publication. The detailed analysis and mechanistic interpretation of the mutations including structural characterization is very good.

We thank the reviewer for the nice overall comment on our work.